# Platelet versus fresh frozen plasma transfusion for coagulopathy in cardiac surgery patients

Jake V. Hinton[1]*, Calvin M. Fletcher[2], Luke A. Perry[3,4], Noah Greifer[5], Jessica N. Hinton[6], Jenni Williams-Spence[7], Reny Segal[3,4], Julian A. Smith[8,9], Christopher M. Reid[7,10], Laurence Weinberg[1,4], Rinaldo Bellomo[4,11,12,13]

1 Department of Anaesthesia, Austin Health, Heidelberg, Australia, 2 Department of Anaesthesiology and Perioperative Medicine, The Alfred Hospital, Melbourne, Australia, 3 Department of Anaesthesia and Pain Management, Royal Melbourne Hospital, Parkville, Australia, 4 Department of Critical Care, University of Melbourne, Parkville, Australia, 5 Harvard University Institute for Quantitative Social Science, Cambridge, MA, United States of America, 6 Department of Orthopaedics, Northern Health, Epping, Australia, 7 Department of Epidemiology and Preventive Medicine, Monash University, Melbourne, Victoria, Australia, 8 Department of Surgery (School of Clinical Sciences at Monash Health), Monash University, Melbourne, Australia, 9 Department of Cardiothoracic Surgery, Monash Health, Clayton, Australia, 10 School of Public Health, Curtin University, Perth, WA, Australia, 11 Department of Intensive Care, Royal Melbourne Hospital, Melbourne, Australia, 12 Department of Intensive Care, Austin Hospital, Melbourne, Victoria, Australia, 13 Australian and New Zealand Intensive Care Research Centre, Monash University, Melbourne, Australia

* jvhinton1@gmail.com

**Data Availability Statement:** All relevant data are within the paper and its Supporting Information files.

## Abstract

### Background

Platelets (PLTS) and fresh frozen plasma (FFP) are often transfused in cardiac surgery patients for perioperative bleeding. Their relative effectiveness is unknown.

### Methods

We conducted an entropy-weighted retrospective cohort study using the Australian and New Zealand Society of Cardiac and Thoracic Surgeons National Cardiac Surgery Database. All adults undergoing cardiac surgery between 2005–2021 across 58 sites were included. The primary outcome was operative mortality.

### Results

Of 174,796 eligible patients, 15,360 (8.79%) received PLTS in the absence of FFP and 6,189 (3.54%) patients received FFP in the absence of PLTS. The median cumulative dose was 1 unit of pooled platelets (IQR 1 to 3) and 2 units of FFP (IQR 0 to 4) respectively. After entropy weighting to achieve balanced cohorts, FFP was associated with increased perioperative (Risk Ratio [RR], 1.63; 95% Confidence Interval [CI], 1.40 to 1.91; P<0.001) and 1-year (RR, 1.50; 95% CI, 1.32 to 1.71; P<0.001) mortality. FFP was associated with increased rates of 4-hour chest drain tube output (Adjusted mean difference in ml, 28.37; 95% CI, 19.35 to 37.38; P<0.001), AKI (RR, 1.13; 95% CI, 1.01 to 1.27; P = 0.033) and readmission to ICU (RR, 1.24; 95% CI, 1.09 to 1.42; P = 0.001).

**Funding:** The author(s) received no specific funding for this work.

**Competing interests:** The authors have declared that no competing interests exist.

## Conclusion

In perioperative bleeding in cardiac surgery patient, platelets are associated with a relative mortality benefit over FFP. This information can be used by clinicians in their choice of pro-coagulant therapy in this setting.

## Introduction

Cardiac surgery is a major consumer of blood products globally [1, 2]. In Australia, it is the most significant surgical indication for fresh frozen plasma (FFP) and platelet (PLTS) transfusion [3]. These two blood products theoretically address differing coagulation pathways. FFP contains clotting factors, fibrinogen, and plasma proteins and may contribute to improved coagulation by increasing their concentration in plasma. Conversely, PLTS may contribute to improved coagulation in the setting of thrombocytopenia or suspected abnormal platelet function [4, 5].

In cardiac surgery, cardiopulmonary bypass (CPB) contributes to both clotting factor deficiency and platelet dysfunction [6, 7]. In the undifferentiated, bleeding, cardiac surgery patient, where a surgical cause of bleeding has been excluded, the relative benefit of addressing platelet dysfunction in preference to clotting factor deficiency is unknown. While transfusion guidelines for packed red blood cells (PRBC) do exist and are based on robust evidence, practice guidelines concerning FFP and PLTS are generally lacking [8]. Thus, in practice, a lack of immediate point-of-care testing and clinician preference contribute to wide variation in blood product administration practices [9]. This ambiguity in practice is compounded by variable clinical outcomes for both FFP and PLTS in cardiac surgery [10–12]. For example, FFP may have advantages in addressing a wide range of clotting factor deficiencies and increasing circulatory volume, but it may also be less effective than previously thought in addressing overall bleeding [13]. Conversely, while PLTS have been historically associated with increased mortality and infection risk, newer studies with greater power suggest potential for a protective effect [14, 15].

Accordingly, we used a multicentre database to investigate the association between FFP and PLTS therapy with key outcomes. We hypothesised that, in entropy-weighted cohorts, PLTS would confer benefit over FFP.

## Methods

### Study design and data

This retrospective cohort study was conducted using the Australian and New Zealand Society of Cardiac and Thoracic Surgeons (ANZSCTS) National Cardiac Surgery Database. Between 2005 and 2021, this database captured comprehensive inpatient data on over 170,000 patients across 58 participating public and private hospitals in Australia [16]. 1-year mortality data was derived from the Australian Institute of Health and Welfare's National Death Index, which was available up until 2018. We reported our results according to the Strengthening of Reporting in Observational Studies in Epidemiology (STROBE) Statement.

### Ethics approval

The Human Research Ethics Committee of the Royal Melbourne Hospital, Melbourne, Australia provided ethical approval for this study (RMH80247) on November 15[th] 2021. The

requirement for additional written informed consent for patient information accessed within the ANZSCTS database was waived for this study.

## Eligibility criteria

We included adult patients undergoing cardiac procedures from 2005 through 2021. Only the primary cardiac procedure during an admission was considered. Patients undergoing primary transcatheter procedures and those with missing blood product data were excluded.

## Exposures

We compared patients who received FFP to those who received PLTS, but not both. Blood product administration was measured cumulatively across the intra- and 30-day post-operative period. In Australia, FFP is either a whole blood or apheresis derived product with an allowable volume of 250-310mL and minimum Factor VIII concentration of 0.70 IU/mL [17]. PLTS are provided as leucocyte depleted pooled or apheresis derived products with a minimum specified platelet count of $200 \times 10^9$ [17].

## Outcomes

The primary outcome was operative mortality, defined as either death during the index hospital admission or death during the 30-day postoperative period. Secondary outcomes included return to theatre, return to theatre for bleeding, prolonged inotrope use greater than four hours, acute kidney injury, new postoperative renal replacement therapy, all infection, pneumonia, septicaemia, wound infection, readmission to ICU, ICU length of stay, ventilation time and four-hour chest drain output. All outcomes were measured up to 30 days postoperatively. Outcome definitions can be found in S1 File.

## Covariates

We selected 73 clinically significant covariates spanning patient demographics, cardiac risk factors, cardiac status at time of surgery, preoperative interventions, surgical factors and intraoperative processes, a comprehensive list of which can is described in the Results below. Transfusion of other blood products (PRBC, CRYO, NOVO) were included.

## Statistical analysis

Baseline categorical characteristics were reported as counts and percentages. Continuous variables were reported as means with standard deviations or medians and interquartile ranges where non-normally distributed. Inter-group differences were assessed for significance using the Chi-squared, unpaired t-test or Wilcoxon Signed Rank test.

Missing data (S1 File), defined as less than 20% missingness, were imputed using Multivariate Imputation by Chained Equations using 20 imputations and 40 iterations [18, 19].

Entropy balancing was used to produce weighted patient cohorts accounting for all covariates for causal effect analysis. Entropy balancing is an inverse probability weighting method which guarantees exact balance of covariate means while maximising precision compared to traditional weighting methods [20].

Weighted datasets were pooled and analysed using Rubin's rules. Treatment effects for each outcome were analysed using relative risk regression (binary) or linear regression (continuous) models applied to the weighted cohort. An E-value analysis was performed on the primary outcome to quantify the potential effect of unmeasured confounding [21].

Two-sided significance testing of p less than 0.05 was used. Effect estimates were presented as risk ratios or adjusted mean differences. All statistical analyses were conducted in R. The WeightThem package was used for entropy balancing [22].

We conducted three sensitivity analyses. In the first analysis, we included only patients who underwent a coronary artery bypass (CABG) procedure to the exclusion of all else, to reduce cohort coagulopathy variability. In the second analysis, we included only patients presenting within the 5-year period from January 1, 2017 to December 31, 2021, to account for changes in transfusion practices over time. In this third analysis, we included only patients who were not also exposed to cryoprecipitate transfusion, given cryoprecipitate is a derived plasma blood product.

## Results

### Patterns of use

Of 174,796 eligible patients, 15,360 (8.79%) received PLTS in the absence of FFP, with a median cumulative dose of 1-pooled unit (IQR 1 to 3) and 6,189 (3.54%) patients received FFP in the absence of PLTS, with a median cumulative dose of 2 units (IQR 0 to 4) (Fig 1). Relative annual administration frequencies of PLTS to FFP are shown in Fig 2, which shows a progressive increase in the PLTS to FFP transfusion ratio. Coadministration with PRBC (59.5%, 68.3%) and CRYO (26.9%, 16.6%) was common in both the PLTS and FFP cohorts, respectively. Before entropy balancing, there was a significant difference in patient demographics (Table 1). After entropy balancing, all covariates were balanced (S1 File).

### Cryoprecipitate outcomes

Major clinical outcomes for the weighted cohort are presented in Table 2.

Operative mortality occurred in 455 of 15,360 patients (3.0%) in the PLTS group, and 300 of 6,189 patients (4.8%) in the FFP group (RR, 1.63; 95% CI, 1.40 to 1.91; P<0.001). Annual risk ratios for operative mortality are shown in Fig 3. The E-value for the point estimate was 2.64 for operative mortality. One year mortality occurred in 695 of 15,360 patients (4.5%) in the PLTS group, and 492 of 6,189 patients (7.9%) in the FFP group (RR, 1.50; 95% CI, 1.32 to 1.71; P<0.001).

FFP was associated with increased rates of 4-hour chest drain tube output (AMD (ml), 28.37; 95% CI, 19.35 to 37.38; P<0.001), new postoperative renal replacement therapy (RR, 1.20; 95% CI, 1.03 to 1.41; P = 0.020), AKI (RR, 1.13; 95% CI, 1.01 to 1.27; P = 0.033), wound infection (RR, 1.46; 95% CI, 1.08 to 1.97; P = 0.014) and readmission to ICU (RR, 1.24; 95% CI, 1.09 to 1.42; P = 0.001). However, FFP was also associated with reduced rates of prolonged inotrope use greater than four hours (RR, 0.93; 95% CI, 0.90 to 0.95; P<0.001), reduced rates of return to operating room/theatre for bleeding (RR, 0.85; 95% CI, 0.73 to 0.97; P = 0.020) and pneumonia (RR, 0.84; 95% CI, 0.74 to 0.94; P = 0.004). All other outcomes were not significantly different.

The primary outcomes in our three sensitivity analyses were concordant with the main cohort primary outcome. FFP was associated with increased operative mortality in the CABG-only cohort (RR, 1.71; 95% CI, 128 to 2.27; P<0.001), in the 2017 through 2021 cohort (RR, 1.59; 95% CI, 1.20 to 2.10; P<0.001) and in the cohort not exposed to cryoprecipitate (RR, 1.62; 95% CI, 1.36 to 1.94; P<0.001). Epidemiological, balance, and further outcome data for these subgroups can be found in S1 File.

## Discussion

### Key findings

In a detailed analysis of a large, multicentric, granular, and curated database, after entropy weighting, we found an independent association between FFP and perioperative and 1-year

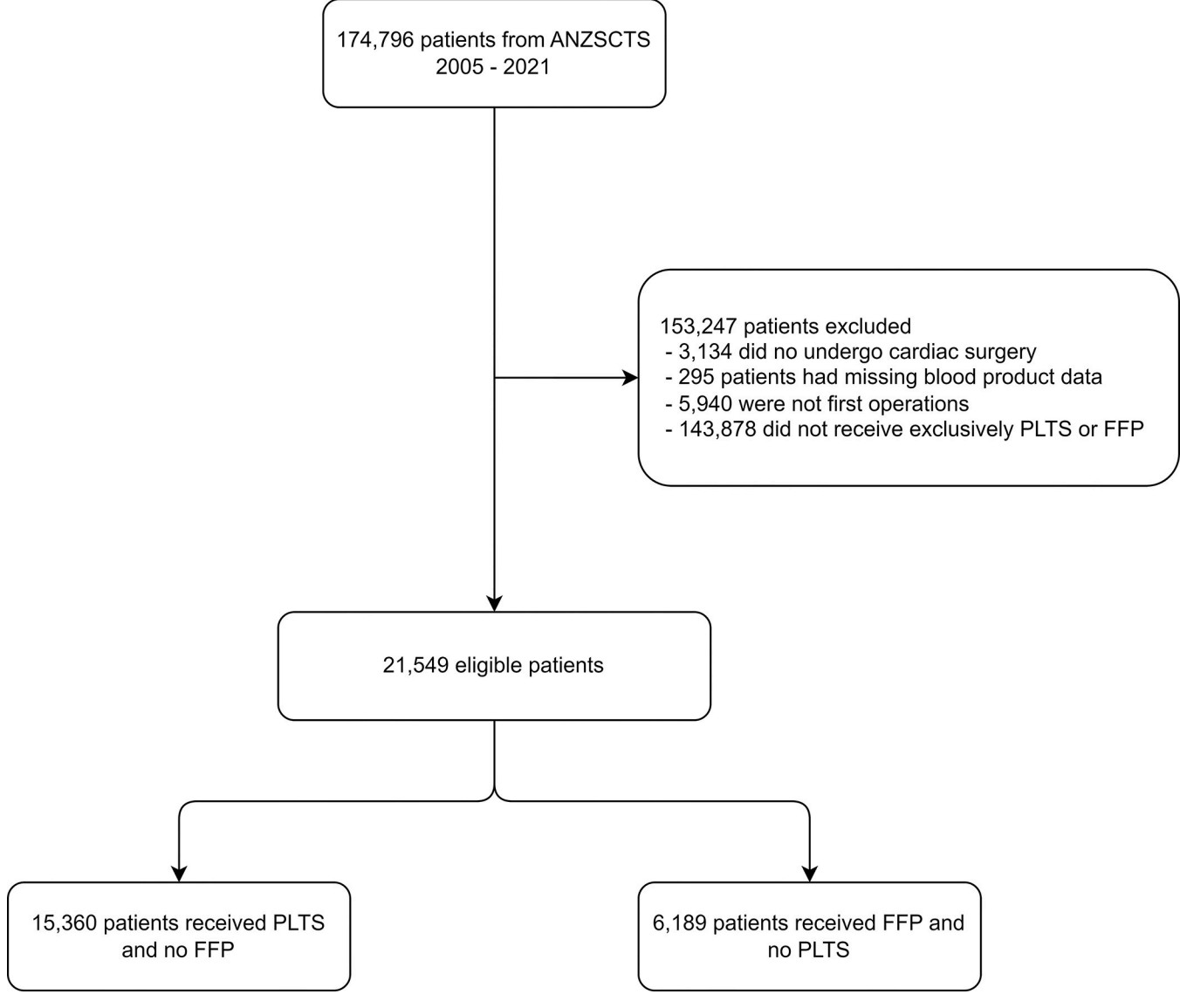

**Fig 1. Patient flow diagram.**

mortality when compared to PLTS. This finding was robust to our sensitivity analyses and to the effects of unmeasured confounding as estimated by E-value analysis.

We also found that PLTS were transfused in the absence of FFP more frequently than vice versa. Moreover, the relative frequency of PLTS to FFP administration has increased significantly between 2005 and 2021. In both cohorts, these blood products were often transfused as an adjunct alongside PRBC. In addition, as expected, PLTS were more often transfused in patients who had recent preoperative aspirin or clopidogrel administration, consistent with their indication for suspected abnormal platelet function. FFP was often transfused in patients requiring preoperative inotrope support. Finally, cross clamp and cardiopulmonary bypass times were significantly longer in the platelet cohort and yet patients who received FFP had higher chest drain output.

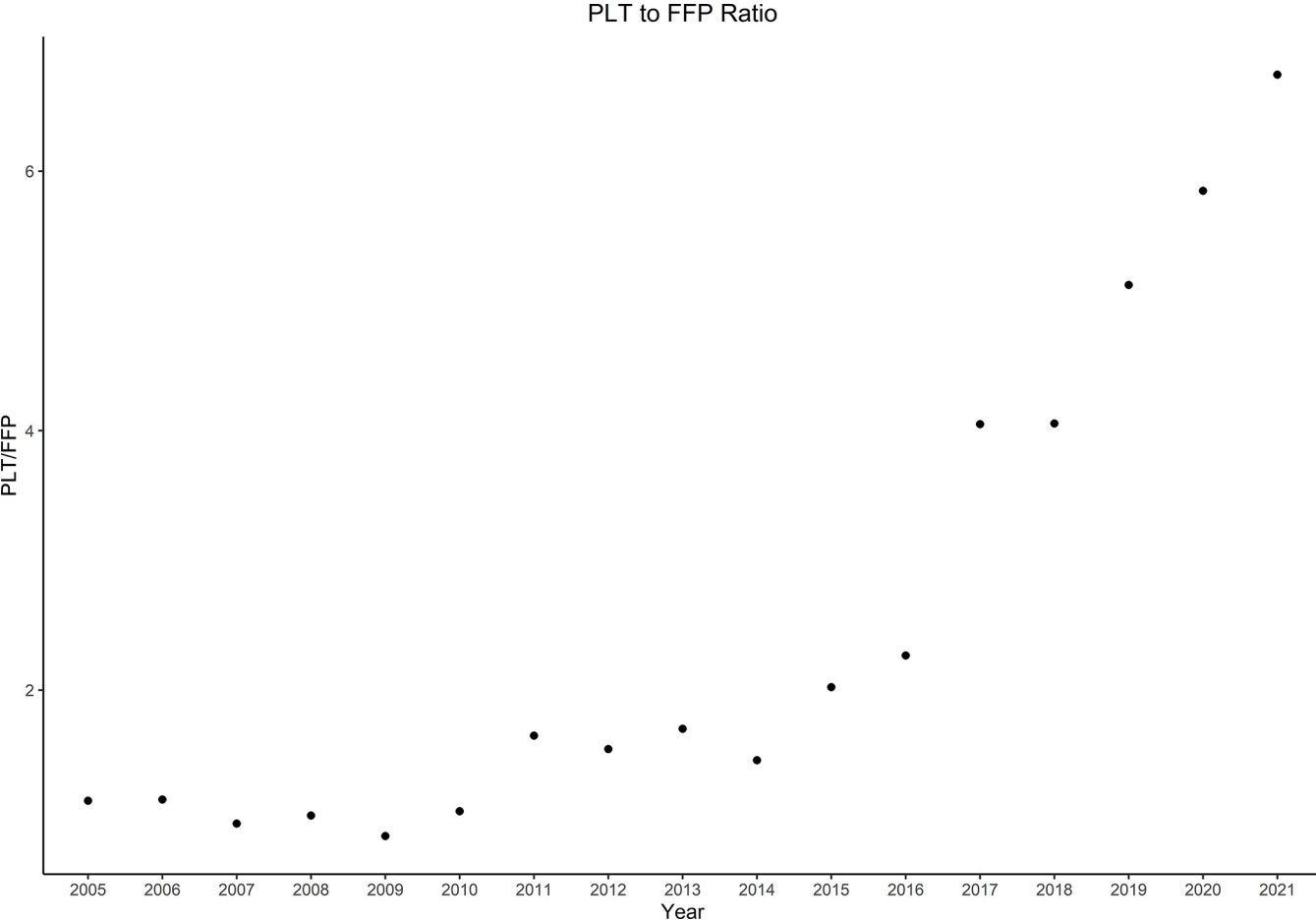

**Fig 2. Ratio of patients receiving platelets to those receiving fresh frozen plasma per year.**

### Relationship to previous studies

Guideline-based indications for FFP and PLTS transfusion are based on the differences in blood product composition and presumed mechanisms because no randomized controlled trials exist to compare these two interventions. However, in practice, FFP and PLTS are frequently transfused according to urgent clinical need and irrespective of either proven coagulopathy or proven platelet dysfunction, respectively [23, 24]. Because of the pressure to intervene rapidly in the face of observable bleeding, the lack of empirical evidence and the paucity of even sufficient evidence behind guidelines, rates of non-adherence to such guidelines are high not only in Australia [24], but also Europe [25] and the USA [26]. Moreover, although modern point-of-care testing technologies theoretically offer the prospect for more targeted blood transfusion practice, they do not yet satisfactorily assess the role of platelets or FFP in haemostasis, require time, are costly, difficult to calibrate, and need operator training. In addition, they have not been shown to alter outcomes in controlled trials and their application delays intervention in a time-pressured environment where visible bleeding drives clinicians to urgent intervention [27]. It is therefore fundamental to provide strong epidemiological data to inform clinician decisions.

To our knowledge, this multicentre retrospective cohort study is the first to report on mortality and other clinically relevant outcomes when comparing FFP and PLTS transfusion in

**Table 1. Baseline characteristics of weighted cohorts.**

| Variable | PLT (n = 15,360) | FFP (n = 6,189) |
|---|---|---|
| **Epidemiology** | | |
| Age (mean (SD)) | 66.59 (12.37) | 66.75 (13.30) |
| BMI (mean (SD)) | 28.25 (5.52) | 27.66 (5.70) |
| Female | 3601 (23.4) | 1814 (29.3) |
| Insurance Status | | |
| Private Health Insurance | 4401 (28.7) | 1531 (24.7) |
| Department of Veteran Affairs | 184 (1.2) | 94 (1.5) |
| Medicare | 10205 (66.4) | 4422 (71.4) |
| Other | 570 (3.7) | 142 (2.3) |
| **Cardiac Risk Factors** | | |
| Smoking | 8651 (56.3) | 3399 (54.9) |
| Diabetes Mellitus | 4322 (28.1) | 1756 (28.4) |
| Hypercholesterolaemia | 10272 (66.9) | 3847 (62.2) |
| Hypertension | 11002 (71.6) | 4389 (70.9) |
| Congestive Heart Failure | 3316 (21.6) | 1741 (28.1) |
| Previous Cardiac Procedure | | |
| CABG | 1113 (7.2) | 377 (6.1) |
| Valve | 1085 (7.1) | 338 (5.5) |
| **Other Comorbidities** | | |
| Cerebrovascular Disease | 1749 (11.4) | 729 (11.8) |
| Peripheral Vascular Disease | 1381 (9.0) | 595 (9.6) |
| Respiratory Disease | 2253 (14.7) | 907 (14.7) |
| Preoperative Dialysis | 483 (3.1) | 122 (2.0) |
| Last Preop Creatinine (median [IQR]) | 88.00 [74.00, 107.00] | 90.00 [75.00, 110.00] |
| **Cardiac Status at Time of Surgery** | | |
| Myocardial Infarction | | |
| Any MI | 5898 (38.4) | 2160 (34.9) |
| Recent MI (< 21 days) | 2021 (13.2) | 884 (14.3) |
| New York Heart Association Class | | |
| I | 5442 (35.4) | 2038 (32.9) |
| II | 5758 (37.5) | 2199 (35.5) |
| III | 3189 (20.8) | 1419 (22.9) |
| IV | 971 (6.3) | 533 (8.6) |
| Canadian Cardiovascular Society Class | | |
| No Angina | 6506 (42.4) | 2708 (43.8) |
| I | 1545 (10.1) | 628 (10.1) |
| II | 3313 (21.6) | 1441 (23.3) |
| III | 2163 (14.1) | 850 (13.7) |
| IV | 1833 (11.9) | 562 (9.1) |
| Number of Diseased Coronary Vessels | | |
| 0 | 4600 (29.9) | 2118 (34.2) |
| 1 | 1274 (8.3) | 486 (7.9) |
| 2 | 2482 (16.2) | 921 (14.9) |
| 3 | 7004 (45.6) | 2664 (43.0) |
| LVEF Estimate | | |
| Normal (>60%) | 7000 (45.6) | 2997 (48.4) |
| Mild (46–60%) | 5004 (32.6) | 1715 (27.7) |

*(Continued)*

**Table 1.** (*Continued*)

| Variable | PLT (n = 15,360) | FFP (n = 6,189) |
|---|---|---|
| Moderate (30–45%) | 2281 (14.9) | 860 (13.9) |
| Severe (<30%) | 792 (5.2) | 466 (7.5) |
| Cardiogenic Shock | 492 (3.2) | 200 (3.2) |
| Arrhythmia | 3077 (20.0) | 1549 (25.0) |
| Infective Endocarditis | 661 (4.3) | 326 (5.3) |
| **Preoperative Interventions** | | |
| Immunosuppressive Therapy (< 30 days) | 537 (3.5) | 170 (2.7) |
| Inotropes (< 24 hours) | 537 (3.5) | 310 (5.0) |
| Nitrates (< 24 hours) | 743 (4.8) | 231 (3.7) |
| Anticoagulation (< 24 hours) | 3729 (24.3) | 1398 (22.6) |
| Steroids (< 24 hours) | 430 (2.8) | 145 (2.3) |
| Aspirin (< 7 days) | 9871 (64.3) | 3160 (51.1) |
| Clopidogrel (< 7 days) | 2189 (14.3) | 523 (8.5) |
| Other antiplatelets (< 7 days) | 939 (6.1) | 356 (5.8) |
| Preop Resuscitation (<1 hour) | 182 (1.2) | 68 (1.1) |
| **Surgical Factors** | | |
| Urgency of Surgery | | |
| Elective | 9066 (59.0) | 3939 (63.6) |
| Urgent | 5033 (32.8) | 1932 (31.2) |
| Emergency | 1209 (7.9) | 297 (4.8) |
| Salvage | 52 (0.3) | 21 (0.3) |
| Type of Surgery | | |
| Isolated CABG | 7383 (48.1) | 2678 (43.3) |
| Isolated Valve Procedure | 2870 (18.7) | 1245 (20.1) |
| Combined CABG/Valve | 2764 (18.0) | 1144 (18.5) |
| Other Cardiac Procedure | 2343 (15.3) | 1122 (18.1) |
| **Intra-/Peri-operative Processes** | | |
| Cardiopulmonary Bypass | | |
| CPB Used | 14926 (97.2) | 5958 (96.3) |
| Cumulative CPB Time (median [IQR]) | 108.00 [79.00, 149.00] | 106.00 [79.00, 142.00] |
| Cumulative Cross Clamp Time (median [IQR]) | 77.00 [53.00, 109.00] | 75.00 [51.00, 107.00] |
| Intra-op Antifibrinolytic Use | 13301 (86.6) | 5334 (86.2) |
| Intra-aortic Balloon Pump | 1061 (6.9) | 308 (5.0) |
| PRBC | | |
| Transfused | 9136 (59.5) | 4225 (68.3) |
| Cumulative Units (median [IQR]) | 1.00 [0.00, 3.00] | 2.00 [0.00, 4.00] |
| CRYO | | |
| Transfused | 4125 (26.9) | 1028 (16.6) |
| Cumulative Units (median [IQR]) | 0.00 [0.00, 2.00] | 0.00 [0.00, 0.00] |
| FFP | | |
| Transfused | 0 (0.0) | 6189 (100.0) |
| Cumulative Units (median [IQR]) | 0.00 [0.00, 0.00] | 2.00 [2.00, 4.00] |
| PLTS | | |
| Transfused | 15360 (100.0) | 0 (0.0) |
| Cumulative Units (median [IQR]) | 1.00 [1.00, 3.00] | 0.00 [0.00, 0.00] |
| NOVO | | |
| Transfused | 97 (0.6) | 22 (0.4) |

(*Continued*)

**Table 1.** (Continued)

| Variable | PLT (n = 15,360) | FFP (n = 6,189) |
|---|---|---|
| Cumulative Units (median [IQR]) | 0.00 [0.00, 0.00] | 0.00 [0.00, 0.00] |

Abbreviations: BMI, body mass index; CABG, coronary artery bypass graft; MI, myocardial infarction; LVEF, left ventricular ejection fraction; CPB, cardiopulmonary bypass; PRBC, packed red blood cells (blood product); CRYO, cryoprecipitate (blood product); FFP, fresh frozen plasma (blood product); PLTS, platelets (blood product); NOVO, NovoSeven; IQR, interquartile range; SD, standard deviation; SMD, standardised mean difference
All values presented as frequency (%) unless otherwise specified.

entropy weighted cohorts of cardiac surgery patients. In general, literature regarding clinical outcomes associated with FFP or PLTS transfusion in cardiac surgery points towards overall increased mortality risk for both [28, 29]. However, a large portion of available evidence is based on studies which are underpowered, do not control for the amount of blood product given, and do not account for the effects of confounding by indication. Finally, recent evidence suggests that the use of PLTS in cardiac surgery patients may be safe [30].

## Secondary outcomes

We did not observe a significant difference in composite infection rates between the FFP and PLTS group. This finding contradicts several previous studies which report reduced infection rates with PLTS and increased infection rates with FFP [29, 30].

**Table 2. Outcomes in the entropy-weighted cohort.**

| Outcomes | PLT (n = 15,360) | FFP (n = 6,189) | RR | CI (95%) | P-Value |
|---|---|---|---|---|---|
| **Primary Outcomes** | | | | | |
| Operative Mortality | 455 (3.0) | 300 (4.8) | 1.63 | (1.40,1.91) | <0.001 |
| **Bleeding Complications** | | | | | |
| Return to Theatre | 1826 (11.9) | 755 (12.2) | 0.98 | (0.90,1.07) | 0.675 |
| Return to Theatre for Bleeding | 848 (5.5) | 293 (4.7) | 0.85 | (0.73,0.97) | 0.020 |
| **Cardiac Complications** | | | | | |
| Prolonged Inotrope Use (>4 hrs) | 9742 (63.4) | 3666 (59.2) | 0.93 | (0.90,0.95) | <0.001 |
| **Fluid Balance Complications** | | | | | |
| AKI | 996 (6.5) | 484 (7.8) | 1.13 | (1.01,1.27) | 0.033 |
| New Postoperative RRT | 552 (3.6) | 286 (4.6) | 1.20 | (1.03,1.41) | 0.020 |
| **Infection** | | | | | |
| All Infection | 1319 (8.6) | 525 (8.5) | 0.91 | (0.82,1.00) | 0.062 |
| Pneumonia | 1111 (7.2) | 400 (6.5) | 0.84 | (0.74,0.94) | 0.004 |
| Septicaemia | 278 (1.8) | 124 (2.0) | 1.02 | (0.81,1.29) | 0.866 |
| Wound Infection | 126 (0.8) | 78 (1.3) | 1.46 | (1.08,1.97) | 0.014 |
| **Hospital Resource Use** | | | | | |
| Readmission to ICU | 692 (4.5) | 391 (6.3) | 1.24 | (1.09,1.42) | 0.001 |
| **Other Outcomes** | | | | | |
| Long Term (1yr) Mortality | 695 (4.5) | 492 (7.9) | 1.50 | (1.32,1.71) | <0.001 |
| **Continuous Outcomes** | | | AMD | CI (95%) | P-Value |
| ICU Length of Stay, hrs (median [IQR]) | 53.00 [37.39, 96.63] | 58.98 [27.43, 114.50] | 2.98 | (-1.51,7.48) | 0.193 |
| Ventilation Time, hrs (median [IQR]) | 12.97 [7.50, 20.86] | 14.75 [9.00, 22.93] | 1.18 | (-1.82,4.18) | 0.441 |
| 4hr Chest Drain Output, ml (median [IQR]) | 240 [150, 400] | 270 [160, 420] | 28.37 | (19.35,37.38) | <0.001 |

Abbreviations: AKI, acute kidney injury; AMD, adjusted mean difference; RRT, renal replacement therapy; ICU, intensive care unit.

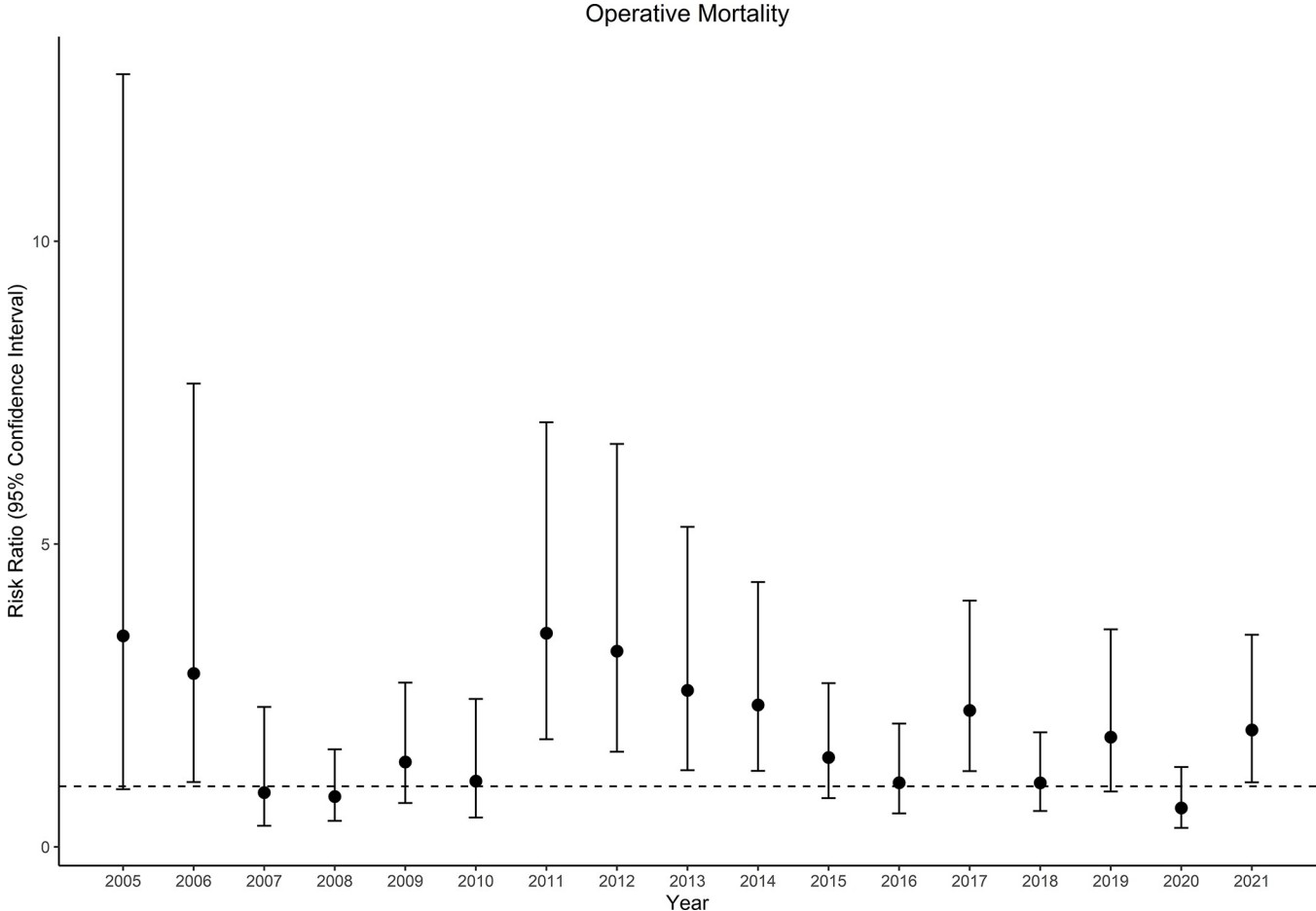

**Fig 3. Operative mortality risk ratio between fresh frozen plasma and platelets per year.**

We observed a positive association between FFP transfusion and new renal replacement therapy. This effect may be explained by the impact of an increase in right atrial pressure with FFP and associated congestion on renal function [31]. We observed a negative association between FFP transfusion and prolonged inotrope use. This association may be related to FFP's transient volume-dependent haemodynamic effects, rather than its ability to prevent ongoing bleeding [32].

FFP was associated with increased rates of readmission to ICU. This finding may be related to fluid redistribution and pulmonary congestion in the days after FFP therapy.

Interestingly, FFP was associated with higher 4-hour chest drain output while PLTS were associated with higher rates of return to theatre for bleeding. FFP transfusion may contribute to a hypervolaemic state and subsequent increased 4-hour chest drain output. While chest drain output may be used clinically as a surrogate for bleeding severity, the decision to return to theatre for resternotomy, on average, occurs 8 hours post admission to ICU postoperatively [33]. As such, the association between 4 hour chest drain output and return to theatre for bleeding is not clear.

## Implications

Our findings imply that the administration of PLTS is likely safe and may be the preferred treatment adjunct over FFP in the management of post-operative bleeding in cardiac surgery

patients. Clinicians should carefully consider the role of FFP in multi-product transfusion protocols and its potential haemodynamic risks. The relative benefit of additional targeted alternate therapies, which more specifically address other clotting factors deficiencies (e.g., cryoprecipitate, fibrinogen concentrate or prothrombin complex concentrate) requires targeted investigations.

## Strengths and limitations

Our study has multiple strengths. First, this is the first large, multi-center study, which directly compares the outcomes associated with FFP and PLTS in cardiac surgery. The patient case-mix and practice across 58 study centres, spanning admissions from 2005–2021 is likely reflective of the whole population. Second, we utilised entropy weighting to create balanced cohorts accounting for over 70 relevant confounders, including the transfusion of other non-FFP, non-PLTS blood products and the preoperative use of antiplatelet agents. Entropy weighting ensured that, statistically, patients had a balanced propensity for receiving FFP or PLTS prior to the actual intervention given, thus emulating a randomized trial situation [34]. Finally, our results were robust to our sensitivity analyses, accounting for differences in operation type, changes in blood transfusion practices over the last five years and the confounding effects of plasma derived cryoprecipitate.

We acknowledge some limitations. First, we were unable to report on the time sequence of events. However, the vast majority of FFP and PLTS transfusions in cardiac surgery occur early in a patient's admission likely before the occurrence of the outcomes of interest [12, 35]. Second, we were unable to report on traditional coagulation tests, point-of-care tests, or platelet counts. However, transfusion practices in cardiac surgery largely remain practitioner dependent. Moreover, the administration of blood products is typically applied rapidly in response observable and clinically significant bleeding. Thus, it occurs before or without coagulation testing, a fact observed worldwide from Australia to Europe and the USA [24–26]. Third, our main cohort study included patients from 2005 to 2021, a period when transfusion practices have changed significantly. However, our sensitivity analysis from 2017 to 2021 was consistent with the main cohort outcomes, supporting the relevance of our findings to modern perioperative practice. Fourth, we were unable to report on transfusion associated adverse reactions. However, such adverse reactions are rare [36]. Fifth, all observational studies have inherent limitations. Thus, our findings may represent the impact of indication bias. For example, PLTS may have been given preferentially in situations of perceived platelet dysfunction and FFP in situations of perceived coagulopathy. However, entropy balancing adjusted for the propensity of receiving PLTS or FFP based on >70 baseline variables, making this a statistically unlikely explanation. Moreover, the potential impact of unmeasured confounders had a high point estimate of 2.64 (E-value) for operative mortality. This implies that such an unmeasured confounder would have to carry an improbably strong impact on mortality and yet have remained unidentified. Finally, despite sophisticated entropy balancing, our observations represent associations only. Thus, they cannot be used to make inferences on causality. However, they are hypothesis-generating, represent best current comparative evidence, and provide the necessary information to justify and design interventional studies to either confirm or refute them.

## Conclusion

When comparing entropy-weighted cohorts of adult patients undergoing cardiac surgery, FFP transfusion was associated with increased perioperative and 1-year mortality relative to PLTS transfusion. These findings support the continued use of PLTS in patients with known or

suspected platelet dysfunction following cardiac surgery and provide the rationale for randomised controlled trials to assess the apparent signal towards harm associated with FFP compared with PLTS administration in cardiac surgery.

## Supporting information

**S1 File. Supplemental information.**
(DOCX)

## Acknowledgments

The ANZSCTS Database thanks all of the investigators, data managers, and institutions that participate in the Program.

## Author Contributions

**Conceptualization:** Jake V. Hinton, Calvin M. Fletcher, Luke A. Perry, Reny Segal, Julian A. Smith, Laurence Weinberg, Rinaldo Bellomo.

**Data curation:** Jake V. Hinton, Calvin M. Fletcher, Luke A. Perry, Jenni Williams-Spence.

**Formal analysis:** Jake V. Hinton, Calvin M. Fletcher, Luke A. Perry, Noah Greifer.

**Investigation:** Jake V. Hinton, Calvin M. Fletcher, Luke A. Perry.

**Methodology:** Jake V. Hinton, Calvin M. Fletcher, Luke A. Perry, Noah Greifer, Rinaldo Bellomo.

**Project administration:** Jake V. Hinton, Calvin M. Fletcher, Luke A. Perry, Laurence Weinberg, Rinaldo Bellomo.

**Resources:** Jake V. Hinton, Calvin M. Fletcher, Luke A. Perry, Jenni Williams-Spence, Christopher M. Reid, Laurence Weinberg, Rinaldo Bellomo.

**Software:** Jake V. Hinton, Calvin M. Fletcher, Luke A. Perry, Jenni Williams-Spence, Christopher M. Reid.

**Supervision:** Luke A. Perry, Reny Segal, Julian A. Smith, Laurence Weinberg, Rinaldo Bellomo.

**Validation:** Jake V. Hinton.

**Visualization:** Jake V. Hinton.

**Writing – original draft:** Jake V. Hinton, Jessica N. Hinton, Rinaldo Bellomo.

**Writing – review & editing:** Jake V. Hinton, Calvin M. Fletcher, Luke A. Perry, Jessica N. Hinton, Reny Segal, Julian A. Smith, Laurence Weinberg, Rinaldo Bellomo.

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
