## [Decision Letter · Decision Letter 0]

29 Nov 2023

PONE-D-23-26662Platelet versus fresh frozen plasma transfusion for coagulopathy in cardiac surgery patientsPLOS ONE

Dear Dr. Hinton,

Thank you for submitting your manuscript to PLOS ONE. After careful consideration, we feel that it has merit but does not fully meet PLOS ONE’s publication criteria as it currently stands. Therefore, we invite you to submit a revised version of the manuscript that addresses the points raised during the review process.

Please revise according to the comments provided by the 2 reviewers.

We look forward to receiving your revised manuscript.

Kind regards,

Luigi La Via

Academic Editor

PLOS ONE

“The authors would like to thank the Australian & New Zealand Society of Cardiac & Thoracic Surgeons National Cardiac Surgery Database Program. The Australian & New Zealand Society of Cardiac & Thoracic Surgeons National Cardiac Surgery Database Program is funded by the Department of Health (Victoria), the Clinical Excellence Commission (New South Wales), Queensland Health (Queensland), and funding from individual Units. Database Research activities are supported through a National Health and Medical Research Council Senior Research Fellowship and Program Grant awarded to Professor Christopher M Reid.”

Reviewers' comments:

Reviewer's Responses to Questions

**Comments to the Author**

1. Is the manuscript technically sound, and do the data support the conclusions?

Reviewer #1: Yes

Reviewer #2: Partly

2. Has the statistical analysis been performed appropriately and rigorously? 

Reviewer #1: Yes

Reviewer #2: I Don't Know

3. Have the authors made all data underlying the findings in their manuscript fully available?

Reviewer #1: Yes

Reviewer #2: Yes

4. Is the manuscript presented in an intelligible fashion and written in standard English?

Reviewer #1: Yes

Reviewer #2: Yes

5. Review Comments to the Author

Reviewer #1: I read with great interest the manuscript by Hinton et al. on the comparison between platelet and fresh frozen plasma transfusion for coagulopathy in cardiac surgery patients. The study is sound and well written. However, there are some minor issues that need to be addressed:

- Line 29. Please replace "nay" with "may".

- In the introduction section, authors should also briefly mention red blood cell transfusion as a possible further strategy in bleeding patients. In fact, in contrast to FFP and PLTS, robust evidence exist on the targets to aim for red blood cell transfusion. Please discuss and cite doi: 10.1053/j.jvca.2023.08.001.

- Why did authors exclude the most recent data from 2022? Please explain.

- Line 65. Why were patients from 2001 to 2005 excluded?

- Line 77-78. Please report the secondary outcomes in the main text, leaving only the definitions in the supplementary material.

- Table 2 is not clear, as no data is provided on the 2 groups, but only the RR. For this reason, it is not possible to determine which group had the best outcomes only by looking at the Table. Please modify.

Reviewer #2: Dear authors,

I have read with interest the manuscript by Hinton et al. entitled « Platelet versus fresh frozen plasma transfusion for coagulopathy in cardiac surgery patients; a multicenter retrospective cohort study »

Here are my commentaries and remarks :

The authors digged out from a very large cardiac database two subgroups of patients who, from a clinical standpoint, diverged from one another.

Indeed, as clinicians, facing perioperative bleeding at the completion of any procedure, the crosstalks between the anesthesiologist and the surgeon, the knowledge of the immediate preoperative patient’s medications and his intrinsic/extrinsic coagulation status, and in some instances, the use of information gathered from point of care tests (ROTEM, Multiplate e.g.) will inevitably lead to different responses in transfusion strategies.

The choice of comparing « Platelets only » and « FFP only » patients intuitively tells us that those groups were not comparable to start with.

Using a newly developed sophisticated statistical method, the authors convincingly states that the model was able to litteraly produce two groups of « comparable patients », close to mimicking a randomized trial of treament A vs B.

Not being a statistician, I personnaly would challenge the authors and request a formal review by a statistical expert on the relevance of the model used herein.

In addition, looking into the details, cryoprecipitate was used in 27% of the patients receiving « platelets only ». Cryoprecipitate being a frozen blood product prepared from fresh frozen plasma, it is rich in plasma proteins such factor 8, fibrinogen, factor 13, von Willebrand factor, and fibronectin.

Instead of perfoming substudies on the CABG subgroup or the most recent surgical group, it would be interesting to exclude from the « platelets only » group all patients having received cryoprecipitate and re-run the statistics.

How the authors can reconcile that the FFP group had higher chest drain output, but that the PLTS group returned more frequently to the OR for bleeding ?

As a detail, in all tables provided, information of the « reference group » for the RR description should be clearly stated, thouth this information can be found in the text.

Large databases often lack granularity secondary to insufficient data registration or missing data at the time of extraction : in the present study, nearly 10% of missing data on the topic of « use of intra-operative antifibrinolytic agent ».

6. PLOS authors have the option to publish the peer review history of their article (what does this mean?). If published, this will include your full peer review and any attached files.

Reviewer #1: No

Reviewer #2: **Yes: **Poncelet Alain

---

## [Author Response · Author response to Decision Letter 0]

1 Dec 2023

PLOS One

Revisions Requested

Suggested Response Document

Editor

Comment 1: Please ensure that your manuscript meets PLOS ONE's style requirements, including those for file naming. 

Response: The manuscript has been updated to comply with style requirements.

Comment 2: We note that you have provided funding information that is not currently declared in your Funding Statement. However, funding information should not appear in the Acknowledgments section or other areas of your manuscript. We will only publish funding information present in the Funding Statement section of the online submission form.

Response: The authors received no specific funding for the completion of this work. We have adjusted the Acknowledgement section to remove all references to funding.

 

Reviewer #1

Comment 1: Line 29. Please replace "nay" with "may".

Response: This has been updated.

Comment 2: In the introduction section, authors should also briefly mention red blood cell transfusion as a possible further strategy in bleeding patients. In fact, in contrast to FFP and PLTS, robust evidence exist on the targets to aim for red blood cell transfusion. Please discuss and cite doi: 10.1053/j.jvca.2023.08.001.

Response: We agree that packed red blood cell (PRBC) transfusion is typically the main strategy in addressing the bleeding patient. While our study is specifically interested in non-PRBC transfusion, we have included the suggested reference with a brief clarification statement in the Introduction as below. 

While transfusion guidelines for packed red blood cells (PRBC) do exist and are based on robust evidence, practice guidelines concerning FFP and PLTS are generally lacking.

Comment 3: Why did authors exclude the most recent data from 2022? Please explain.

Response: Data from 2022 patient admissions is not yet available in the latest version of the ANZSCTS database

Comment 4: Why were patients from 2001 to 2005 excluded?

Response: We have corrected an error in the timeframe stated over which the ANZSCTS database collected patient information. This has been amended to 2005-2021 throughout the manuscript. There is no patient data available prior to 2005.

Comment 5: Please report the secondary outcomes in the main text, leaving only the definitions in the supplementary material.

Response: The secondary outcomes investigated are now stated in the main text as requested.

Comment 6: Table 2 is not clear, as no data is provided on the 2 groups, but only the RR. For this reason, it is not possible to determine which group had the best outcomes only by looking at the Table. Please modify.

Response: Thank you for this comment. Table 2 has been updated to include descriptive statistics regarding cohort outcomes.

 

Reviewer #2

Comment 1: The choice of comparing « Platelets only » and « FFP only » patients intuitively tells us that those groups were not comparable to start with. Using a newly developed sophisticated statistical method, the authors convincingly states that the model was able to litteraly produce two groups of « comparable patients », close to mimicking a randomized trial of treament A vs B. Not being a statistician, I personnaly would challenge the authors and request a formal review by a statistical expert on the relevance of the model used herein.

Response: Thank you for seeking clarification of our statistical methodology. Entropy weighting is a type of Inverse Probability Weighting (IPW) that can be used to create two cohorts with balanced covariate distribution. The statistical methods outlined in our study have been conceptualised and formally reviewed prior to submission by our named author Noah Greifer, who is a data science specialist, statistical consultant and programmer for the Institute for Quantitative Social Science (IQSS) Harvard University, USA.

Comment 2: In addition, looking into the details, cryoprecipitate was used in 27% of the patients receiving « platelets only ». Cryoprecipitate being a frozen blood product prepared from fresh frozen plasma, it is rich in plasma proteins such factor 8, fibrinogen, factor 13, von Willebrand factor, and fibronectin.

Instead of perfoming substudies on the CABG subgroup or the most recent surgical group, it would be interesting to exclude from the « platelets only » group all patients having received cryoprecipitate and re-run the statistics.

Response: Given cohorts were balanced after entropy weighting, any effects of cryoprecipitate transfusion were balanced between cohorts, and thus not significant.

However, we have now also included a sensitivity analysis including only patients who were not exposed to cryoprecipitate for comparison. The results are provided in the revised manuscript and online supplement. The results are concordant with the main cohort study, and an additional comment has been included in the Results section of the manuscript to highlight this, as below.

The primary outcomes in our three sensitivity analyses were concordant with the main cohort primary outcome. FFP was associated with increased operative mortality in the CABG-only cohort (RR, 1.71; 95% CI, 128 to 2.27; P<0.001), in the 2017 through 2021 cohort (RR, 1.59; 95% CI, 1.20 to 2.10; P<0.001) and in the cohort not exposed to cryoprecipitate (RR, 1.62; 95% CI, 1.36 to 1.94; P<0.001). 

Comment 3: How the authors can reconcile that the FFP group had higher chest drain output, but that the PLTS group returned more frequently to the OR for bleeding ?

Response: Thank you for this comment. A statement exploring this interesting topic has been added to the Discussion, shown below.

Interestingly, FFP was associated with higher 4-hour chest drain output while PLTS were associated with higher rates of return to theatre for bleeding. FFP transfusion may contribute to a hypervolaemic state and subsequent increased 4-hour chest drain output. While chest drain output may be used clinically as a surrogate for bleeding severity, the decision to return to theatre for resternotomy, on average, occurs 8 hours post admission to ICU postoperatively.[33] As such, the association between 4 hour chest drain output and return to theatre for bleeding is not clear.

Comment 4: As a detail, in all tables provided, information of the « reference group » for the RR description should be clearly stated, thouth this information can be found in the text.

Response: Please see response to Reviewer 1, Comment 6

Comment 5: Large databases often lack granularity secondary to insufficient data registration or missing data at the time of extraction : in the present study, nearly 10% of missing data on the topic of « use of intra-operative antifibrinolytic agent ».

Response: I believe the full comment from the reviewer has been cut off. However, we have provided justification for our strategy in relation to missing data below.

The ANZSCTS database is one of the world’s largest, most granular, prospectively collated cardiothoracic surgery databases. All missing data included in the study were assumed Missing at Random, and imputed using Multivariate Imputation by Chained Equations. Current literature suggests that Multivariate Imputation reduces bias in Missing at Random data, even with large degrees of missingness. A citation for this (https://doi.org/10.1016/j.jclinepi.2019.02.016) has been included in the references of the manuscript.

Missing data (S1 File), defined as less than 20% missingness, were imputed using Multivariate Imputation by Chained Equations using 20 imputations and 40 iterations.[18, 19]

---

## [Decision Letter · Decision Letter 1]

18 Dec 2023

Platelet versus fresh frozen plasma transfusion for coagulopathy in cardiac surgery patients

PONE-D-23-26662R1

Dear Dr. Hinton,

We’re pleased to inform you that your manuscript has been judged scientifically suitable for publication and will be formally accepted for publication once it meets all outstanding technical requirements.

Kind regards,

Luigi La Via

Academic Editor

PLOS ONE

Additional Editor Comments (optional):

Reviewers' comments:

Reviewer's Responses to Questions

**Comments to the Author**

1. If the authors have adequately addressed your comments raised in a previous round of review and you feel that this manuscript is now acceptable for publication, you may indicate that here to bypass the “Comments to the Author” section, enter your conflict of interest statement in the “Confidential to Editor” section, and submit your "Accept" recommendation.

Reviewer #1: All comments have been addressed

Reviewer #2: All comments have been addressed

2. Is the manuscript technically sound, and do the data support the conclusions?

Reviewer #1: Yes

Reviewer #2: Yes

3. Has the statistical analysis been performed appropriately and rigorously? 

Reviewer #1: Yes

Reviewer #2: I Don't Know

4. Have the authors made all data underlying the findings in their manuscript fully available?

Reviewer #1: Yes

Reviewer #2: Yes

5. Is the manuscript presented in an intelligible fashion and written in standard English?

Reviewer #1: Yes

Reviewer #2: Yes

6. Review Comments to the Author

Reviewer #1: The authors successfully addressed all the comments provided. I believe the manuscript can be accepted in its present form.

Reviewer #2: (No Response)

7. PLOS authors have the option to publish the peer review history of their article (what does this mean?). If published, this will include your full peer review and any attached files.

Reviewer #1: No

Reviewer #2: **Yes: **Poncelet Alain

---

## [Editor Report · Acceptance letter]

5 Jan 2024

PONE-D-23-26662R1 

PLOS ONE

Dear Dr. Hinton, 

I'm pleased to inform you that your manuscript has been deemed suitable for publication in PLOS ONE. Congratulations! Your manuscript is now being handed over to our production team.

Kind regards, 

on behalf of

Dr. Luigi La Via 

Academic Editor

PLOS ONE